# Development and Validation of an Observational Game Analysis Tool with Artificial Intelligence for Handball: Handball.ai

**DOI:** 10.3390/s23156714

**Published:** 2023-07-27

**Authors:** Moises Marquina, Demetrio Lozano, Carlos García-Sánchez, Sergio Sánchez-López, Alfonso de la Rubia

**Affiliations:** 1Deporte y Entrenamiento Research Group, Departamento de Deportes, Facultad de Ciencias de la Actividad Física y del Deporte (INEF), Universidad Politécnica de Madrid, C/Martín Fierro 7, 28040 Madrid, Spain; moises.mnieto@upm.es (M.M.); alfonso.delarubia@upm.es (A.d.l.R.); 2Health Sciences Faculty, Universidad San Jorge, Autov A23 km 299, Villanueva de Gállego, 50830 Zaragoza, Spain; dlozano@usj.es; 3SPORT Research Group (CTS-1024), Centro de Evaluación y Rehabilitación Neuropsicológica, University of Almería, Ctra. Sacramento, s/n, 04120 Almería, Spain; ssl832@inlumine.ual.es

**Keywords:** handball, performance indicators, artificial intelligence

## Abstract

Performance analysis based on artificial intelligence together with game-related statistical models aims to provide relevant information before, during and after a competition. Due to the evaluation of handball performance focusing mainly on the result and not on the analysis of the dynamics of the game pace through artificial intelligence, the aim of this study was to design and validate a specific handball instrument based on real-time observational methodology capable of identifying, quantifying, classifying and relating individual and collective tactical behaviours during the game. First, an instrument validation by an expert panel was performed. Ten experts answered a questionnaire regarding the relevance and appropriateness of each variable presented. Subsequently, data were validated by two observers (1.5 and 2 years of handball observational analysis experience) recruited to analyse a Champions League match. Instrument validity showed a high accordance degree among experts (Cohen’s kappa index (k) = 0.889). For both automatic and manual variables, a very good intra- ((automatic: Cronbach’s alpha (α) = 0.984; intra-class correlation coefficient (ICC) = 0.970; k = 0.917) (manual: α = 0.959; ICC = 0.923; k = 0.858)) and inter-observer ((automatic: α = 0.976; ICC = 0.961; k = 0.874) (manual: α = 0.959; ICC = 0.923; k = 0.831) consistency and reliability was found. These results show a high degree of instrument validity, reliability and accuracy providing handball coaches, analysts, and researchers a novel tool to improve handball performance.

## 1. Introduction

The analysis and evaluation of sports performance is one of the research areas that has become increasingly relevant in the study of team sports, especially in collaborative—opposition sports [1,2,3]. To this effect, ad hoc tools have been developed with the operational objective of providing greater information to stakeholders and practitioners to optimise sports performance [4]. Many of these analysis instruments are based on the concepts of ‘observational methodology’ [5] and ‘notational analysis’ [6] which, through the collection, analysis and interpretation of data, aim to draw empirical conclusions by assessing the context and dynamics of the sport [7]. This methodology, characterised by being a flexible and rigorous procedure [8], aims to contribute to a deeper analysis of the dynamic reality of the game by understanding the determining performance factors, as well as the detection of evolutionary trends [9].

The efficient use of observational instruments for game analysis requires two essential conditions: validity and reliability. While the former evaluates the adequacy and appropriateness of an item or aspect [8], the latter refers to the reproducibility of the values of a test or other measurement in repeated trials on the same individuals [10]. Reliability tests in sports performance analysis are conducted to ensure objectivity in the data collection process through inter-observer agreement [11]. Nevertheless, concordance within the same observer is also necessary to guarantee data quality. Thus, an observational instrument should allow high-reliability values, both between observers (inter-observer) as well as in the different observations of the same subject (intra-observer), and it is necessary to report these data quality processes [9].

Studies in collaborative/opposition team sports based on the use of observational methodology have multiplied exponentially. Consequently, the design and validation of ad hoc observational instruments have become common and frequent in elite sports, aimed at identifying performance factors [12]. One of the sports where the production has been most prolific is football. Thus, in the systematic review by Clemente et al. [13], 21 studies were identified, of which 8 focused on tactical tests (e.g., FUT-SAT), 10 on technical tests (e.g., Loughborough Soccer Passing Test) and 3 on tests concurrent with physical aspects (e.g., MT5M). In other sports (e.g., basketball) there is an amalgam of instruments with different purposes, such as the influence of contextual variables on tactical thinking [14], the player discrimination by position according to physical–technical execution patterns [15] or the relevance of physiological parameters [16]. In other sports such as rugby [17], water polo [18] and ice hockey [19], tools based on observational methodology have also been developed.

Regarding handball, the scientific evidence shows a clear orientation towards the design of observational instruments for technical–tactical game analysis, mainly at the offensive level [20]. Some examples focused on the effectiveness of offensive game systems [21] and offensive tactical behaviour in critical phases [22], on the reliability of the register in the counterattack phase [23], on attacking effectiveness according to game and contextual variables [24], on the effectiveness of offensive actions according to gender [25], on the evaluation of individual technical–tactical performance in competition [26] or on offensive performance in elite handball [27,28]. Specifically, numerous investigations based on observational methodology at the offensive level focused on the analysis of the throwing action with the aim of evaluating the specific weight of each playing position in the performance [29,30,31,32,33,34,35]. Moreover, due to the regulatory changes introduced with regard to the ‘empty goal’ rule [36], more and more authors are concerned with designing valid and reliable tools to analyse 7 × 6 situations [37]. Thus, these kind of records are in full expansion phase in modalities such as beach handball [38,39] or associated with the combined analysis of the physical and physiological demands of players [40] and referees [41] using polar coordinates.

One of the fastest-growing research and development areas in handball is performance analysis based on artificial intelligence—AI [42]. The adoption of AI and statistical models, based on game-related statistics obtained through notational analysis and observational methodology [43], has arisen with the clear objective of providing information before, during and after the competition [42]. One of the most pursued study objects has been team discrimination—winners and losers. Research has been carried out in different sport contexts, such as the Olympic Games [44,45,46], World Championships [47,48,49,50,51], European Championships [52], Pan-American Championships [53], European Champions League [54,55], national leagues [56,57], semi-professional or amateur contexts [58,59] and youth competitions [60]. Other investigations based on game-related statistics focused on the role played by the goalkeeper [61], the impact of the ‘empty goal’ rule of substitution of the goalkeeper by a court player [62], the influence of variables associated with sport experience [63], the establishment of an individual performance index rating—’Play Score’ [64]—the defence-attack relationship [65], the relevance of contextual variables [66] or coaching decisions (e.g., time-outs) [67].

To the best of our knowledge, no handball performance analysis software has been able to determine in depth the impact of factors affecting the dynamics of game pace in handball (e.g., number of 1 × 1 actions that are successfully resolved by the offense team) beyond the outcome of the actions (i.e., goal or no goal), homogenising the tracking, automatically generating statistical models in situ and classifying players according to specific performance parameters (e.g., number of times that a player/team was involved in the possession of the ball, in general and in the different phases of the game). Therefore, and because of the frequent use of observational methodology for technical–tactical game analysis in handball according to performance factors and the present relevance of AI based on game-related statistics in sport, the objective of this study was to design and validate a recording instrument based on observational methodology (hereafter, Handball.ai) capable of identifying, quantifying, classifying and relating individual and collective tactical behaviours in handball, ensuring compliance with the principles of validity, reliability, accuracy and generalisability. The authors hypothesise that Handball.ai will further analyse competitive handball performance according to automatic time and event control (i.e., number of possessions per player and per team), instantaneously generate statistics based on mathematical models independently of the tracker’s intervention (i.e., player score) and complete simplified event tracking based only on the team in possession of the ball.

## 2. Materials and Methods

A nomothetic observational, follow-up and multidimensional design was used for the present study [8]: nomothetic because there is a sample of several teams; follow-up because several championships are analysed over time and compared with each other; and multidimensional because different dimensions are taken into account. This N/S/M design leads to a series of decisions about the sample, the observation-recording instruments and the analysis procedure.

The process begins with data capture, which can be accomplished through different means, such as video cameras, sensors or tracking devices. These data are collected during sporting events and provide valuable information on athlete performance, tactics used, game dynamics and other relevant aspects. Once the data are collected, AI comes into play. Machine learning algorithms and natural language processing are used to analyse and extract meaningful information from observational data. These algorithms are pre-trained using sample data sets containing labelled examples, allowing the AI to learn patterns and trends from the data [68].

Therefore, the AI in the Handball.ai tool can perform a wide variety of tasks in sports observational analysis. For example, it can automatically identify and follow players during a match, detect movement patterns and analyse the tactics and strategies used by teams. It can also perform advanced statistical analysis, such as calculating the average performance of a player in different situations or comparing the performance of different teams. In addition, the AI can provide visualisations and graphical representations of the data, making it easier for coaches, players and fans to understand and analyse. These visualisations can include motion graphs, heat maps, tactical diagrams and other visual elements that help reveal hidden patterns and trends in observational data.

### 2.1. Procedures

The design and validation of the instrument followed a four-stage process (Figure 1). The first phase of the research was a literature review that allowed the definition of the theoretical framework and the procedures to be carried out and the determination of the study design [69]. 

The second phase was to define the variables in a criteria system and category analysis. Descriptors were established, characterising the behaviours and classifying them, based on the scientific literature, into different categories and variables (Table 1).

The third phase was the validation of the instrument. The content validity was determined by expert judgement. First, a questionnaire was created. Ten experts of different nationalities participated in the validation process.

An invitation explaining the context, description and purpose of the study, together with a guide to assess the following aspects of the components and their levels, was sent via email: The experts assessed the “wording” and “suitability” sections of each item, using a quantitative Lickert-type scale from 1 to 10. In addition, they made an overall qualitative assessment of each element regarding whether they considered it appropriate, where they expressed their alternatives to certain aspects that they would personally improve. To do so, they recorded: (a) the degree to which each item belongs to the object of study (“suitability”). A quantitative score was given by each expert to each unit of the instrument to determine the relevance with which each item should form part of the observation instrument. (b) Degree of precision and correctness (wording): by means of using quantitative scores given by the experts, the degree to which each item was correctly worded and defined was observed. (c) Degree of qualitative comprehension: the reflections provided by the experts on the different questions of the instrument were collected. 

Subsequently, the data were transformed to a 0.1–1 scale, averaging the experts’ responses for each criterion and category, according to factor. All the categories of the instrument presented agreement and acceptance values above 0.8 and were accepted [70].

Once the instrument had been validated, a fourth and final phase was carried out to validate the data. For this purpose, the observers were trained and made several observations of the same match to achieve data reliability using Cohen’s kappa concordance coefficient [71], as indicated by several studies [58,72].

### 2.2. Participants

To validate the instrument (third phase), 10 experts of different nationalities participated in the validation process. The sample who participated to validate the instrument had to meet the following inclusion criteria established in the research: (I) qualification: coach with Master Coach level certified by the EHF (European Handball Federation) or Level III EHF Rinck Convention; (II) years of experience: have been training for more than 10 years in national and international categories; and (III) current season: be training in the national category in the current season [73].

For data validation (fourth phase), two observers were recruited. These expert observers had Level III EHF Rinck Convention certification with 1.5 and 2 years of experience, respectively, in analysing handball using observational software. Both observers were of different nationalities (mean age: 36 years) and had experience as coaches for 13 ± 4 years. A training process of the observers lasted 21 days. 

The match selected for validation corresponded to the Champions League round of 16 match between Telekom Veszprém HC and OTP Bank PICK Szeged, played on 30 March 2023. To analyse the match, data were accessed using the match recording made by the broadcasting rights company according to the data collection criteria [74].

To define the study sample, the following inclusion criteria were used: (a) the match to be analysed had to belong to an international competition. Because this type of competition has a high level of competitiveness, to access this championship, teams must qualify in their respective national leagues and then pass a preliminary phase that only allows the teams classified in the first positions to continue. For this reason, a match of the 2022/2023 Champions League was used.

### 2.3. Observation Tool

An ad hoc software was designed to record the individual actions of players during a handball match in real time. This software allows working with the multidimensional observation system and was named Handball.ai™ (Version: 1.2.4 (Heidelberg, Germany)). This software allows the collection of match data through a graphical interface containing two windows, one showing the video of the match and the other a graphical representation of the players and their numbers and the different team systems. The software is presented on a screen where different buttons appear to record each variable. A mobile device with a touch screen (tablet) was used to record all the events in real time as quickly and easily as possible (Figure 2).

The following variables were defined and described to clarify data collection. The variables that define the individual actions of the players were divided into groups (Table 1): period, team, event, phase, offense system, defence system (Figure 3), offense player position, defence player position, shooter position (Figure 4), shot location (Figure 5) and possession.

Within the different dimensions, some of the variables are calculated manually, i.e., by the observer clicking. On the other hand, there are other variables calculated automatically by the software. This calculation is performed by means of automatic calculation parameters, i.e., an automatic calculation based on manually marked events (Table 2).

### 2.4. Data Validation

The inter-observer reliability assessment process was similar to methods used in previous research [58,75]. The onboarding of the observers consisted of the following phases: (1) meeting with the platform developer to get to know the platform and become familiar with it; (2) from 1 week to 10 days for the observers to test it and ask questions about the process; (3) a 2-h session, normally separated by 1 h, where they observed a match together with the platform developer and recorded the observed data simultaneously; and (4) a session made up of the analysis of 2 complete live matches, comparing the results. In each training session the observers were subjected to the same conditions: (i) the observer was isolated in a room to maintain the intra-sessional connection; and (ii), observations were made at the same time and in the same place under stable conditions and without the presence of any person who might interfere directly or indirectly in the process, except the researcher.

Once the training period was over, observation was carried out to validate the data. The same match was recorded by the two observers, and the results were compared between them (inter-observer). One week later, the same match was observed again by the same observers, and the results were compared with the previous analysis (intra-observer).

The reliability and internal consistency of the data collected were analysed. To determine the case proportion, Cohen’s kappa index (k) was used, which analyses the agreement degree between observers after excluding the proportion of cases in which the agreement between them is the result of chance [76]. Kappa values can range from −1.0 to 1.0. Agreement in the interpretation of the kappa value was assessed as follows: <0 no agreement; 0.01–0.20 poor agreement; 0.21–0.40 fair agreement; 0.41–0.60 moderate agreement; 0.61–0.80 good agreement; and 0.81–0.99 very good agreement [77]. Cronbach’s alpha (α) and intraclass correlation coefficients (ICC) were used to determine internal consistency. Internal consistency indices and reliability thresholds ranged from 0 to 1 [78] and were set at: for α (internal consistency), < 0.50 unacceptable, 0.51–0.60 poor, 0.61–0.70 questionable, 0.71–0.80 acceptable, 0.81–0.90 good and ≥ 0.91 excellent [79]; for ICC (reliability), ≤ 0.50 poor, 0.51–0.75 moderate, 0.76–0.90 good and ≥ 0.91 excellent [80].

## 3. Results

Cohen’s kappa index (k) was used to determine the degree of agreement among the experts. A Cohen’s kappa concordance was achieved of very good agreement at 0.889 (Table 3). The analysis of variance (one-factor ANOVA) regarding variables of ranges with a normal distribution (Table 3) did not show significant differences between the averages of the different experts (0.101 f; 1.000 sig). 

Table 4 shows the mean intra-observer and inter-observer reliability and internal consistency tests of the automatic variable data collected by the Handball.ai software. Both can be considered very good at 0.957 and 0.937, respectively.

Table 5 shows the means of the intra-observer and inter-observer reliability and internal consistency tests for the data of the manual variables collected by the observers. Both can be considered very good at 0.913 and 0.904, respectively.

## 4. Discussion

In this study we analysed the validity and the inter- and intra-observer reliability of Handball.ai in identifying, quantifying, classifying and relating individual and collective tactical behaviours in handball. The main results obtained in the reliability tests (ICC and Cohen’s kappa) and internal consistency tests (Cronbach’s alpha and ICC), intra-observer and inter-observer, demonstrate that Handball.ai is a valid, reliable and precise tool to register individual and collective tactical behaviours in handball. In particular, the data of the automatic variables collected by Handball.ai (ICC = 0.957 and 0.937) and the data of the manual variables collected by the observers (ICC = 0.913 and 0.904) can be considered very good. Therefore, this reveals that the system is a very reliable software capable of registering automatic and manual variables, serving as an observational instrument to analyse elite handball.

The expert panel has played an important role in improving and clarifying the definition, relevance and appropriateness of each variable used in Handball.ai, as in similar studies developed in other sports [17,81,82]. Specifically, the Cohen’s kappa index (κ = 0.889) showed a very good agreement between experts in order to validate each variable of the instrument. In addition, one of the main strengths of the expert panel was that the 10 coaches have experience at international competitions and are aware of the observational methodology.

Handball.ai contains some dimensions and variables that have been previously validated in other instruments and that serve to contextualise the situation of the game in which the event takes place, such as the period of the match, the phase of the game, the offense system or the defence system [21,22,23,24,26,28]. Some of these variables (e.g., period or offense system) are recorded automatically by Handball.ai, which is the first handball observation instrument that uses the advantages of AI [41,42,69]. This innovation is a great advance with respect to the software developed and validated until now [83,84], as it will reduce the time required to record and code the events during the match.

In addition, the instrument includes other variables to expand the information related to a shot on goal, such as shooter position and shot location. These variables have also been widely used in other observation instruments that have been validated [22,23,24,26,27,29]. Likewise, Handball.ai is a valid and reliable tool to register game events that have already been considered decisive for achieving maximum performance in handball, such as goals, saves, turnovers, technical mistakes, assists, etc. [43,44,45,46,47,48,49,50,51,52,53,54,55,56,57,58,59].

Therefore, this instrument incorporates new variables that had not been included to date in an observation tool with similar characteristics [21,22,23,24,26,28]. One of these variables is the number of possessions, an aspect that will allow us to analyse in depth the dynamics of game pace in competition. It also includes another new variable that allows us to quantify the number of 1 × 1 actions that are successfully resolved by the offensive team. This variable represents a new advance in handball game analysis, as it will allow us to know exactly which player has generated the offensive superiority, regardless of whether the action ends in a goal or 7 m. This information was not considered in other observation instruments [21,22,23,24,26,28] and will be especially useful for handball coaches and performance analysts.

It would be appropriate to note that although Handball.ai has shown good validity and reliability both intra- and inter-observer in elite handball (0.913–0.957 and 0.904–0.937), we should be careful with the use of this software in other handball modalities, such as beach handball and wheelchair handball.

Future studies could evaluate the generalisation of the system with a larger sample of matches in similar or different competitions, such as national teams’ championships or national league tournaments.

## 5. Conclusions

The findings reported in this study demonstrate that Handball.ai is a valid, reliable, and precise tool for identifying, quantifying, classifying and relating individual and collective tactical behaviours in elite handball competitions. In particular, the main results obtained in the reliability tests and internal consistency, both intra-observer and inter-observer, demonstrate that the data of the automatic and manual variables (collected by the Handball.ai and by the observers, respectively) can be considered very good.

Furthermore, this instrument presents two main advantages compared to other software validated to date. Firstly, the use of AI to collect automatic variables will reduce the time required to record and code the events during the match. Secondly, this instrument incorporates new variables, such as number of possessions and number of 1 × 1 actions that are successfully resolved by the offensive team. These variables will allow coaches to perform a deeper analysis of handball.

Therefore, handball coaches and performance analysts should consider the high reliability and internal consistency of the automatic and manual variables to conduct analysis during elite competitions. Specifically, this tool would help technical staff to: (i) design better training programs focused on improving the decisive technical actions of the game; (ii) make comprehensive analyses of the technical–tactical performance (individual and collective) of their opponents; (iii) develop a more effective game plan; and (iv) improve talent identification and player recruitment. In addition, data collected by Handball.ai might be used by researchers for the development of new research in performance analysis in handball.

There is no doubt that AI will continue to transform analysis and evaluation of elite sports. Consequently, further research is needed to analyse other AI applications, such as predicting match outcomes, assisting coaches on line-ups, tactics and player rotations, supporting referee decisions during the competition and understanding the value of players from an economic perspective.

## Figures and Tables

**Figure 1 sensors-23-06714-f001:**
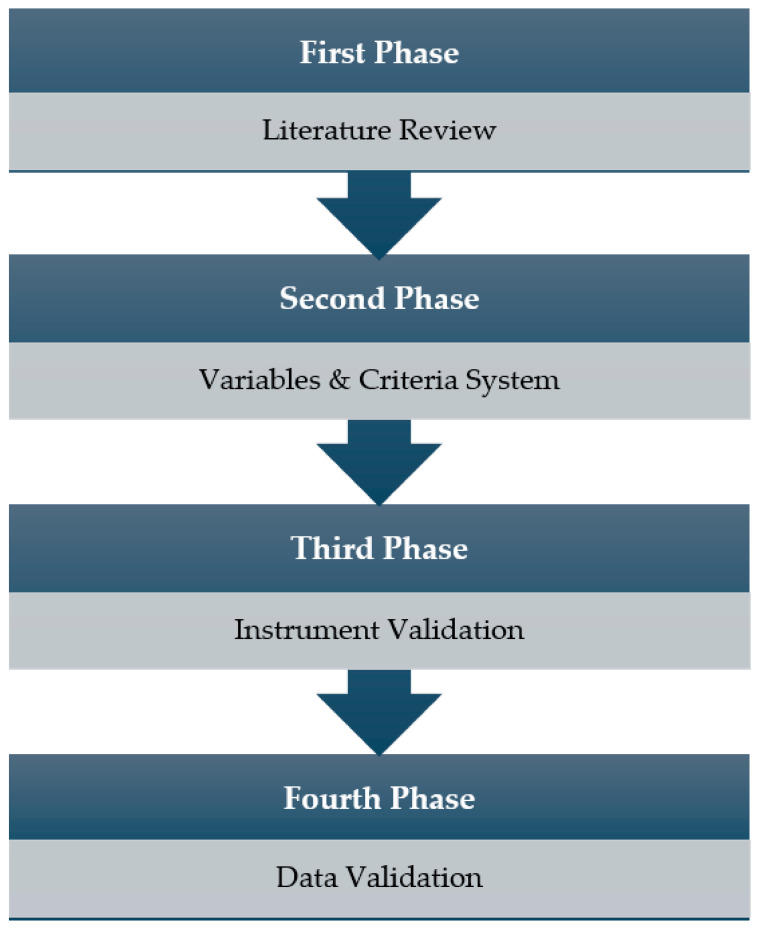
Structure of different phases of the validation process.

**Figure 2 sensors-23-06714-f002:**
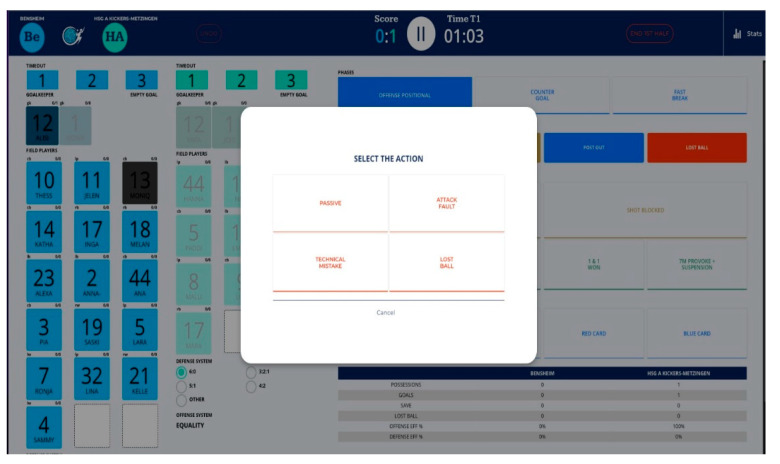
Buttons of the different actions of Handball.ai.

**Figure 3 sensors-23-06714-f003:**
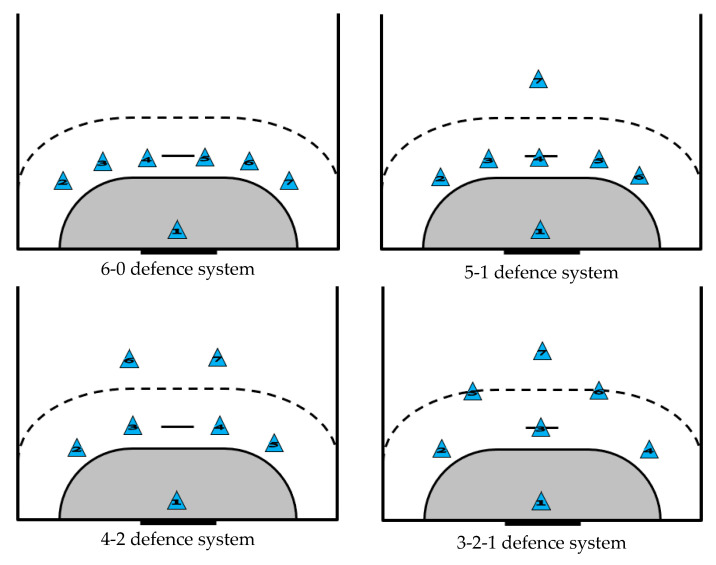
Graphic representation of the different defence systems.

**Figure 4 sensors-23-06714-f004:**
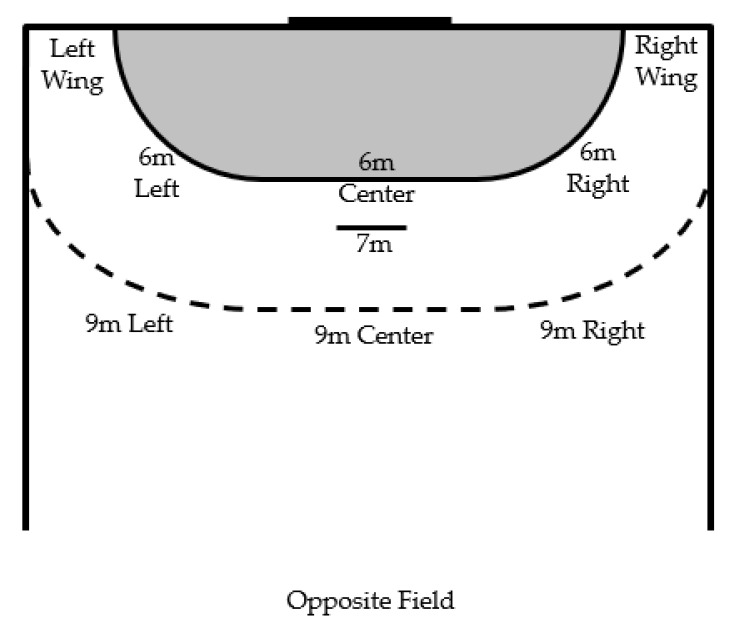
Graphic representation of shooter position.

**Figure 5 sensors-23-06714-f005:**
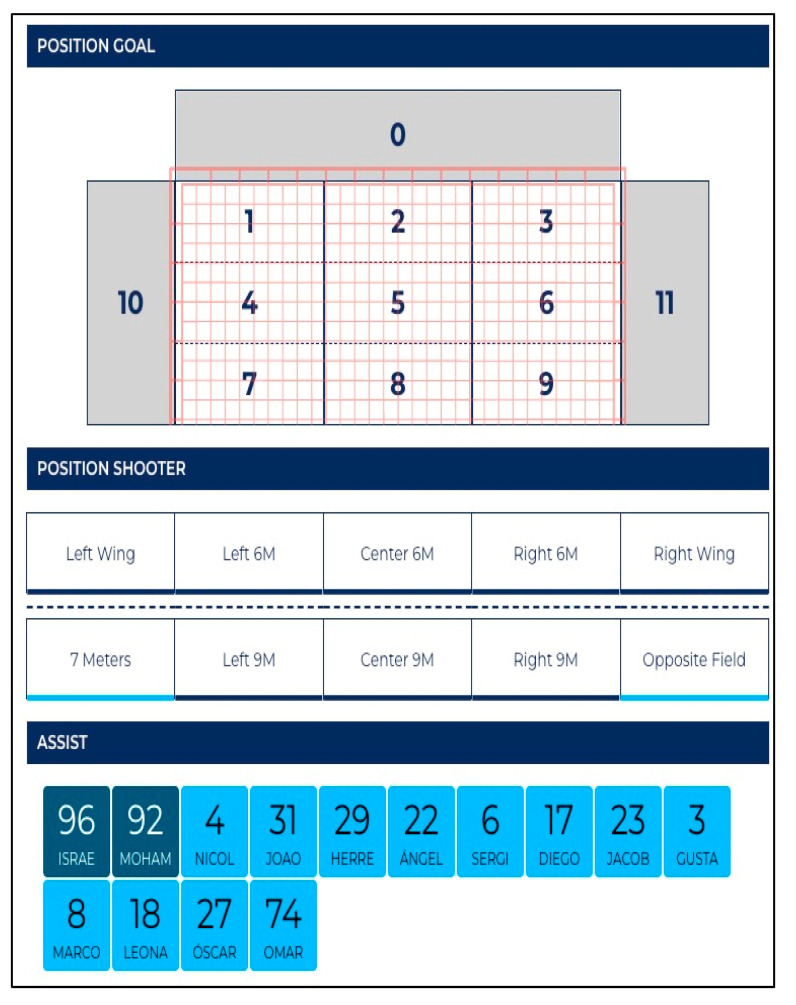
Graphic representation of shot location.

**Table 1 sensors-23-06714-t001:** Definitions of the game-related statistics.

Dimension	Variable	Definition
Period	First Half	Division of the time duration of a handball match. Comprises the first and second half.
	Second Half	
Team	Name	Name of each of the contending clubs or teams in the match.
Event	Goal	Moment that one team with the possession of the ball gets the ball into the opponent’s goal.
	Save	Moment that the goalkeeper stops the ball from a shot by the opposing team.
	Post Out	Moment that one team with the possession of the ball realizes one shot, with the finality to score a goal, but the shot goes out or against the post.
	Attack Fault	Infraction committed by a player as a result of a foul on a defender indicated by the referee.
	Lost Ball	The attacking team loses ball possession, and the defending team takes ball possession.
	Technical Mistake	Loss of ball possession through a rule infringement, e.g., steps, double dribble.
	Passive	Infraction indicated by the referee if the team in ball possession does not change its attacking behaviour or does not take a throw-in after the referee has indicated the passive play warning.
	Fault	Stopping of an opposing player by the player in ball possession without generating any suspension (yellow card, red card, blue card and 2 min).
	Shot Blocked	Technical action of a defending player who intercepts the trajectory of the ball in a throwing attempt on goal.
	2 min provoked	Player in ball possession attempts to score a goal and is illegally prevented from doing so by another player on defence, incurring a 2 min suspension.
	7 m provoked	Player in ball possession tries to score a goal and another player on defence stops him, illegally interrupting a clear goal-scoring opportunity.
	7 m provoked + 2 min suspension	Player in ball possession tries to score a goal and another player in defence stops him illegally, interrupting a clear goal-scoring opportunity and incurring a 2 min suspension.
	1 × 1 Won	Player in ball possession overtakes the player on defence, generating a clear goal-scoring opportunity or a 7 m provoked, 2 min suspension and 7 m provoked + 2 min suspension.
	2 min	Exclusion of a player for 2 min indicated by the referee due to repeated fouls, unsporting conduct, an incorrect change or as a result of a disqualification.
	Yellow Card	Disciplinary sanction awarded by the referee to a player and/or coaching staff member as a result of a fault or unsporting behaviour.
	Red Card	Disciplinary sanction awarded by the referee to a player and/or coaching staff member for severe misconduct or for receiving a 2 min suspension three times.
	Blue Card	Disciplinary sanction imposed by the referee on a player and/or coaching staff member as a consequence of serious misconduct or misconduct involving a written report.
Phase	Positional Offense	Chance to score a goal by the attacking team against a structured defence. Action of attacking teams playing with structured attack in the opponent’s field.
	Fastbreak	Recovery of ball possession by the defence due to an event (e.g., turnover) and this team searches for a clear action to score a goal against an unstructured defence.
	Countergoal	Recovery of ball possession by the defence due to a goal, and this team searches for a clear action to score a goal against an unstructured defence.
Defence System		Structure that both teams have in any moment of defending time during handball game.
Offense System		Structure that both teams have in any moment of attacking time during handball game.
Offense Player Position		Playing position occupied by an offensive player on the court
Defence Player Position		Playing position occupied by a defence player on the court
Shooter Position		On the field (left wing, 6 m left, 6 m centre, 6 m right, right wing, 9 m left, 9 m centre, 9 m right, opposite field and 7 meters).
Shot Location		(In the goal) with a difference in number 1, 2, 3 (from left to right up); 4, 5, 6 (from left to right in the middle); and 7,8,9 (from left to right down).
Possession		Number of times that a player/team was involved in the possession of the ball, in general and in the different phases of the game (positional offense, countergoal and fastbreak).

**Table 2 sensors-23-06714-t002:** Dimensions analysed and their calculation method.

Dimension	Tracking
Period	Automatic
Team	Automatic
Offense System	Automatic
Offense Player Position	Automatic
Defence Player Position	Automatic
Possession	Automatic
Duration	Automatic
Players	Automatic
Event	Manual
Phase	Manual
Defence System	Manual
Shooter Position	Manual
Shot Location	Manual

**Table 3 sensors-23-06714-t003:** Validation data experts panel.

	Expert 1	Expert 2	Expert 3	Expert 4	Expert 5	Expert 6	Expert 7	Expert 8	Expert 9	Expert 10
Variables	κ	κ	κ	κ	κ	κ	κ	κ	κ	κ
Period	1	1	1	1	1	1	1	1	1	1
Team	1	1	1	1	1	1	1	1	1	1
Offense System	1	0.78	1	0.78	1	0.8	1	0.78	1	0.82
Offense Player Position	0.79	0.64	0.82	0.64	0.78	0.78	0.64	0.64	0.78	0.78
Defence Player Position	0.81	0.78	0.83	0.78	0.82	0.82	0.78	0.78	0.82	0.82
Possession	1	1	1	1	1	1	1	1	1	1
Duration	0.8	0.82	0.79	0.82	0.78	0.82	0.82	0.82	0.82	0.82
Players	1	1	1	1	1	1	1	1	1	1
Event	1	1	1	1	0.82	1	0.82	1	0.82	1
Phase	0.8	0.78	0.61	0.78	0.78	0.82	0.78	0.78	0.78	0.82
Defence System	1	1	1	1	1	1	1	1	1	1
Shooter Position	0.79	0.82	0.78	0.82	0.82	0.78	0.82	0.82	0.78	0.78
Shot Location	0.81	0.78	0.77	0.78	0.8	0.8	0.8	0.78	0.8	0.8
Mean	0.908	0.877	0.892	0.877	0.892	0.894	0.882	0.877	0.892	0.895
Total	0.889									
	ANOVA (Sig) 1.000 F (0.101)									

Note: κ: Cohen’s kappa. Sig: Significance. F: Variance.

**Table 4 sensors-23-06714-t004:** Validity coefficients for intra- and inter-observer internal consistency (Cronbach’s alpha) and reliability (intra-class correlation coefficients and Cohen’s kappa) for automatic variables.

	Intra-Observer			Inter-Observer		
Variable Group	α	ICC (95%)	κ	α	ICC (95%)	κ
Period	1	1	1	0.995	0.989	0.929
Team	0.951	0.907	0.908	0.989	0.978	0.919
Duration	-	0.977	0.955	0.895	0.845	-
Offense System	0.986	0.972	0.98	0.992	0.984	0.91
Offense Player Position	0.982	0.965	0.958	0.984	0.969	0.915
Possession	1	1	0.976	1	1	0.958
Team Players		-	0.645	-	-	0.61
Mean	0.984	0.970	0.917	0.976	0.961	0.874
		0.957			0.937	

Note: α: Cronbach’s alpha; ICC: Intra-class correlation coefficients; κ: Cohen’s kappa.

**Table 5 sensors-23-06714-t005:** Validity coefficients for intra- and inter-observer internal consistency (Cronbach’s alpha) and reliability (intra-class correlation coefficients and Cohen’s kappa) for manual variables.

	Intra-Observer			Inter-Observer		
Variable Group	α	ICC (95%)	κ	α	ICC (95%)	κ
Event	0.996	0.993	0.942	1	0.999	0.955
Phase	0.891	0.803	0.83	0.909	0.833	0.814
Shooter Position	0.945	0.895	0.821	0.981	0.963	0.601
Defence Player Position	0.984	0.968	0.797	0.992	0.984	0.873
Shot Location	0.977	0.955	0.901	0.912	0.838	0.913
Mean	0.959	0.923	0.858	0.959	0.923	0.831
		0.913			0.904	

Note: α: Cronbach’s alpha; ICC: Intra-class correlation coefficients; κ: Cohen’s kappa.

## Data Availability

Not applicable.

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
