# Peer review of "Development and Validation of an Observational Game Analysis Tool with Artificial Intelligence for Handball: Handball.ai"

_sensors, 2023, doi:10.3390/s23156714_

Round 1

Reviewer 1 Report

First of all, congratulations on the research you have carried out. You present a novel, well-structured study that meets the minimum quality requirements to be published in this journal. 

The introduction has all the information related to creating a field of knowledge so that the reader can better understand the research, the results and the discussion and conclusions are very well presented and show the great knowledge that the authors have about the sport on which the research revolves, handball. 

The only doubt I have about this work is the sample with which the research was carried out; could the authors explain why they only watched one match, and do they not think that the data quality analysis could have been completed by carrying out a generalizability study and obtaining a minimum number of observation sessions to be able to generalize accurately?

If the authors can explain this in the article or complete it with the suggestion, I think it is an article with a lot of potential to be published. 

Reviewer 2 Report

Dear authors,

About your paper titled: Development and validation of an observational game analysis tool with artificial intelligence for handball. Handball.ai

First, thank you for the opportunity to review this work. Your word had a practical application for coaches. However, I have some short questions linked to the scientific strength/justification of your research. 

After reading and reviewing your paper, follow my brief reviews:

Introduction 

Could you please provide the originality level or your research gap? It was not clear during the background development. You must let clear: Which aspects of your study differ from previous research? During your discussion, you made it clear in lines 320-324: To our knowledge, this instrument incorporates new variables that had not been included to date in an observation tool of similar characteristics [21-24, 26, 28]. One of these variables is the number of possessions, an aspect that will allow us to analyze in depth the dynamics of game pace in competition. It also includes another new variable that allows us to quantify the number of 1x1 actions that are successfully resolved by the offense team.

Is it possible to develop research hypotheses after making the state of art knowledge clear? If yes, please add it.

Conclusion

You can conclude more specifically, not also saying that Handball.ai is a valid, reliable, and precise tool for identifying, quantifying, classifying, and relating individual and collective tactical behaviors in elite handball competitions; but, in really, briefly relating this tool’s new advantages.  

After these arrangements, we can deeply identify your research novelty level.

No further suggestions

Best regards

 Minor editing of English language required

Reviewer 3 Report

Although the article is very interesting and suitable for any reader in the topic of game analysis with artificial intelligence, several recommendations should be considered for publication:

Relevant comments:

1.       Abstract: identify the problem of analysis and evaluation of sports and improve the importance of the investigation.

2.       The research model and hypothesis should be described in detail in the Methodology section. Evaluate the possibility of including a methodology outline with the different investigation phases.

3.       To increase the understanding of the manuscript, the section “2.2. procedures” should be before the introduction section 2.

4.       Discussion: Evaluate the possibility of creating a comparative study of the results obtained. The discussion must be a quantitative study.

5.       The conclusions are basic. Improve this point. Include the highlights of the results obtained.

6.       The manuscript structure is complicated and needs to be modified.

7.       Evaluate the possibility of introducing an appendix with additional information. It will allow a better understanding of the paper.

8.       Include a One-Way Analysis of Variance (ANOVA) to identify if there are any statistical differences between the means of experts.

9.       In opinion of this reviewer, a brief comment about the future of analysis and evaluation of sports with AI should be included in the conclusions.

10.   I don´t agree with the validation of instrument. The opinion analysis of 10 experts is not sufficient to validate the instrument. Include other instrument to validate the software.

11.   In conclusion, the goal of this manuscript is to promote the software Handball.ai, but there is not sufficient scientific method to accept it for publication in its current version.

Minor comments:

1.       There are format mistakes.

Round 2

Reviewer 3 Report

The authors have taken into account all the reviewers' comments. The quality and scientific interest of the article has been improved. In the opinion of this reviewer, the article can be published in its current version.